# Roles of Three *FgPel* Genes in the Development and Pathogenicity Regulation of *Fusarium graminearum*

**DOI:** 10.3390/jof10100666

**Published:** 2024-09-24

**Authors:** Lu Cai, Xiao Xu, Ye Dong, Yingying Jin, Younes M. Rashad, Dongfang Ma, Aiguo Gu

**Affiliations:** 1Key Laboratory of Sustainable Crop Production in the Middle Reaches of the Yangtze River, College of Agriculture, Yangtze University, Jingzhou 434025, China; 2023710834@yangtzeu.edu.cn (L.C.); xuxiao0913@foxmail.com (X.X.); 2021720806@yangtzeu.edu.cn (Y.D.); 202072787@yangtzeu.edu.cn (Y.J.); 2Jiangsu Academy of Agricultural Sciences, Jiangsu Coastal Area Institute of Agricultural Sciences, Yancheng 224000, China; 3Plant Protection and Biomolecular Diagnosis Department, Arid Lands Cultivation Research Institute (ALCRI), City of Scientific Research and Technological Applications (SRTA-City), New Borg El-Arab 21934, Egypt; younesrashad@yahoo.com; 4Jiangsu Product Quality Testing & Inspection Institute, Nanjing 210007, China; nj180618@163.com

**Keywords:** *Fusarium graminearum*, pectin lyase, gene knockout, pathogenicity

## Abstract

Fusarium head blight (FHB) is a devastating fungal disease caused by *Fusarium graminearum*. Pectin lyase, a pectinase, acts on the α-1,4-glycosidic linkage of galacturonic acid primarily by β-elimination. In this study, three pectin lyase genes (*FgPel1*, *2*, *3*) in *F. graminearum* were selected, and deletion mutants (Δ*FgPel1*, *2*, *3*) were constructed by homologous recombination for functional characterization. The gene deletions affected the morphology and growth rate of *F. graminearum* on pectin medium at various concentrations, with the growth rate of Δ*FgPel1* being more significant. The growth of Δ*FgPel1* colonies slowed at pH 4, with optimal growth at pH 6.5, whereas Δ*FgPel2* and Δ*FgPel3* exhibited greater inhibition at pH 8. Colony morphology and diameter of the deletion mutants showed no significant differences compared to the wild-type strain PH-1, and there was no effect on conidial production or germination rate. Pathogenicity assays demonstrated that gene deletion significantly reduced the ability of *F. graminearum* to infest corn silks and wheat ears, and that Δ*FgPel2* showed a more pronounced reduction in pathogenicity on wheat spikes. In summary, the pectin lyase genes (*FgPel1*, *2*, *3*) are involved in pectin utilization and are influenced by external pH conditions, which attenuate the pathogenicity of *F. graminearum* without affecting its vegetative growth or asexual spore formation. These findings elucidate the roles of these genes and provide a basis for controlling FHB.

## 1. Introduction

Fusarium head blight (FHB) is one of the most significant diseases of wheat. *F. graminearum* Schwabe is an important pathogenic fungus that causes FHB [1,2,3]. *F. graminearum* colonizes plant surfaces and spreads through wind and rain in warm and humid conditions. The mycelium invades the internal tissues of wheat ears and grows within the cells, leading to cell death [4,5]. In addition to causing significant yield and quality losses, FHB results in the production of various toxins in wheat kernels, such as deoxynivalenol (DON) and zearalenone (ZEA). These toxins, in addition to those produced by other *Fusarium* species, can contaminate grains used in animal feed and human food, causing a significant risk to both animals and humans [6,7]. Chemical fungicides, due to their high efficiency and broad application range, remain one of the most effective methods for controlling wheat blight [8,9]. However, the overuse of chemical fungicides in recent years has led to increased pathogen resistance. Therefore, it is crucial to use chemical agents rationally and to develop highly effective and environmentally friendly biofungicides [10].

Plant cell walls are a primary barrier that protects plants from attacking pathogens and play a crucial role in the interactions between plants and microorganisms [11,12]. The major components of plant cell walls are cellulose, hemicellulose, and pectin. Pectin is primarily found in dicotyledons, and is less abundant in grasses [13,14,15,16]. Pectin is a versatile plant polysaccharide. Four different substructures have been identified in pectin: high galacturonic acid (HG), xylogalacturonic acid (XGA), rhamnogalacturonic acid I (RGI), and rhamnogalacturonic acid II (RGII) [17]. Pectin lyases (PL, EC 4.2.2.2) degrade pectin polymers by the β-elimination mechanism, resulting in the formation of 4,5-unsaturated oligogalacturonides [18,19]. Pectin lyases are classified based on their substrate specificity and mode of action into several classes, including pectin methylesterases, hydrolases, and pectin lyases. Pectin lytic enzymes are widespread in fungi and bacteria, with most being produced by fungi [20]. In filamentous fungi, pectin lytic enzymes (PELs) are produced by different genera such as *Aspergillus*, *Penicillium*, and *Fusarium*. Deletion of the *MoPL1* gene encoding a pectin lyase in *Magnaporthe oryzae*, the causal agent of rice blast, resulted in a significant reduction in the pathogen virulence [21]. In *A. flavus*, which infects cotton, the *pecA* gene encodes the PG-active P2c that facilitates infection, and deletion of this gene significantly reduced the cotton boll damage [22]. *Verticillium dahliae*, a broad-spectrum fungal pathogen, has a pectinolytic enzyme-encoding gene (*VdPEL1*), and the mutant *VdPEL1rec*, which lacks the enzyme activity, loses its ability to induce cell death and plant resistance [23]. Simultaneous knockout of *pg1* and *xry1*, which encode cell wall degradation enzymes in wheat pathogens, resulted in a significant reduction in the virulence of *F. graminearum* [24]. Targeted mutation of the *frpl* gene, encoding an F-box protein in *F. oxysporum f.* sp. *lycopersici*, resulted in a decreased expression of pectinase (*PL1*, *PG1*, and *PG2*) and xylanase (*XYL2* and *XYL5*) genes, impairing the ability of the pathogen to colonize and invade tomato roots [25]. CUT2, a keratinase in *M. oryzae*, is critical for the invasion process, with expression increasing during cell maturation and invasion. Mutants deficient in CUT2 showed abnormalities in their pathogenicity, conidial formation, and hyphal morphology [26].

In this study, we selected three pectin-degrading genes from *F. graminearum*: *FgPel1*, *FgPel2*, and *FgPel3*. We constructed a deletion mutant for these genes and analyzed its phenotype. The results showed that deletion of these genes did not affect the vegetative growth of *F. graminearum*, but impaired its ability to utilize pectin. In addition, mycelial growth was affected by external pH levels. Pathogenicity experiments showed that gene deletion reduced the ability of the fungus to infect corn stalks and wheat ears. Understanding this pathogenicity mechanism can help us to understand the plant responses to fungal infections and develop appropriate control strategies.

## 2. Materials and Methods

### 2.1. Fungal Strains, Media, and Culture Conditions

Wild-type *F. graminearum* PH-1 and gene deletion mutants were inoculated on potato sucrose agar medium (PSA) (200 g of potato, 20 g of sucrose, 15 g of agar, and 1 L of water). PSA medium with different pH and different concentrations of pectin was used to test the mycelial growth of the strains. The strains were inoculated in sodium carboxymethyl cellulose liquid medium (CMC), which consisted of yeast extract (1 g), magnesium sulfate heptahydrate (1 g), potassium phosphate (1 g), ammonium nitrate (1 g), sodium carboxymethylcellulose (15 g), and water to a final volume of 1 L. This medium was used for cultivating the strains to produce conidia. The yeast extract peptone dextrose liquid medium (YEPD) (dextrose 20 g, peptone 10 g, yeast extract 3 g, and 1 L of water) was employed for the examination of the conidial morphology, as detailed in reference [27].

### 2.2. Sequential Analysis

The protein sequences of the three genes (GenBank registry numbers: NC_026475.1, NC_026474.1, and NC_026476.1) were downloaded from the NCBI database and subsequently visualized. The selected protein sequences were analyzed using the online software SMART 9.0 (http://smart.embl-heidelberg.de (accessed on 23 February 2024)) and Pfam 35.0 (http://pfam.xfam.org/search (accessed on 23 February 2024)) to identify their structural domains, which were then visualized using IBS 2.0. The predicted amino acid sequences were compared and used to construct the phylogenetic tree using the neighbor-joining (NJ) method, which was implemented in MEGA 11.0 software [28,29].

### 2.3. Phylogenetic Tree Construction and Physicochemical Characterization of FgPels

The 3-D structure of the pectin lyase protein was predicted by the I-TASSER5.0 server (https://zhanggroup.org/I-TASSER/ (accessed on 1 April 2024)). I-TASSER (Iterative Threading Assembly Refinement) is a hierarchical approach for protein structure prediction and a structure-based functional annotation [30]. TheSAVES v6.0 server (https://saves.mbi.ucla.edu// (accessed on 5 April 2024)) and the PROSA-web server were used to generate Ramachandran, ERRAT, and Verify3D plots for validating the quality of the model structure [31,32]. The physicochemical properties of the *FgPels* proteins were characterized using the ProtParam tool of the EXpASY server. The output data from this server included the amino acid composition, theoretical isoelectric point (pI), molecular weight, instability index, aliphaticity index, and the grand average of hydropathicity (GRAVY) [33].

### 2.4. Generation of the Knockout Mutants

The split-marker approach was used to generate the gene replacement constructs for the *FgPel* genes. The genomic DNA of PH-1 was used as a template, and the upstream fragment L1 and downstream fragment L2 of the target gene were amplified using primers *FgPel*-1F/*FgPel*-2R and *FgPel*-3F/*FgPel*-4R, respectively. The fusion fragments LH1 and LH2 were amplified using the primer pairs *FgPel*-1F/HY-R and YG-F/*FgPel*-4R, respectively, with L1 and H1 as template 1 and L2 and H2 as template 2. The product purification kits (FastPure Gel DNA Extraction Mini Kit) were used to recover the fusion fragments [34]. The PEG-mediated protoplast transformation method was used for the transformation of the protoplasts. After the transformation was completed, the transformants were screened for HPH resistance using the primer pairs *FgPel*-5F/*FgPel*-6R, *FgPel*-7F/H855R, H856F/*FgPel*-8R, and H850/H852. It was determined that the target genes were absent [35]. The primer pairs used are listed in Appendix A.

### 2.5. qRT-PCR Assays

Mycelia from PH-1, *FgPel1*, *FgPel2*, and *FgPel3* strains, as well as wheat spikes infected by PH-1 at different time points, were ground in liquid nitrogen. RNA was extracted using the Trizol method, and each RNA sample was subjected to reverse transcription. The expression levels of each gene were measured by quantitative real-time PCR using the primers listed in Appendix A. Each experiment was independently repeated three times.

### 2.6. Effect of Variation in pH and Pectin Concentration on Mycelial Growth

The use of PSA plates allowed for the growth of PH-1 and mutant strains, which were incubated in the dark at 25 °C for three days. After this incubation period, the fungal discs were placed in the center of 20 mL PSA plates of pH values of 4, 6.5, and 8, with the front side of the discs in contact with the culture medium. Three replicates of each experiment were conducted and incubated in the dark at 25 °C for three days. Thereafter, the diameters of the colonies were measured using vernier calipers by the criss-cross method, and photographs were taken. The experiment was repeated three times.

The mycelial plug was removed and inoculated onto 20 mL of pectin agar medium at concentrations of 0.1, 0.5, and 1%, with the front side of the plug in contact with the medium. The cultures were incubated in the dark at 25 °C for six days. The colony diameters were then measured and photographed using the criss-cross method. The experiment was repeated three times.

### 2.7. Asexual Reproduction of Mutants

To study the effects of gene deletion on mycelial growth, conidial production, and morphology of *F. graminearum*, PH-1 and mutant strains were activated on PSA medium. Five replicates were prepared on 9 cm dishes with 20 mL of PSA medium, placing the colonies centrally with their fronts in even contact with the dish surface. Colony diameters were measured after 3 days of incubation at 25 °C, with plates inverted during incubation [36]. The mycelium at the colony edge was scraped off with a sterile toothpick, and a 6 mm punch was used to extract a fungal plug, which was then inoculated into 50 mL of CMC medium. One fungal plug was placed in each bottle, and the bottles were placed in a shaker at 150 rpm. The fungal cultivation was conducted at 25 °C in a shaker at 150 rpm for five days, with three replicates for each strain. After 5 days, the spore solution was filtered through a sterile filter paper. Then, 10 μL of the filtered solution was aspirated and counted with a hemocytometer, and the process was repeated 10 times. The spore morphology was observed and photographed using a fluorescence microscope (Nikon DS-Ri2, Tokyo, Japan). The spore solution was then centrifuged at 6000 rpm for 5 min, and the supernatant was discarded. The spores from the precipitate were transferred to 100 mL of YEPD and incubated on a shaker at 25 °C and 150 rpm. They were observed and photographed under a microscope at 6 h [37].

### 2.8. Determination of Pathogenicity on Wheat at Flowering Stage and Pathogenicity Experiment of Corn Silk Inoculation

The PH-1 and mutant strains were inoculated in CMC medium. The conidia concentration was adjusted to 1 × 10^5^ spores/mL and inoculated at the wheat flowering stage by injecting 10 µL of the spore suspension into grains in the middle and lower middle of the wheat spikelets. An equal amount of water was inoculated as a control (ck). After inoculation, the bag was sprayed with water to maintain moisture for 2 days and then removed. After 14 days, the incidence of wheat ear infection was counted and recorded with a camera [38]. The disease index is calculated as follows:
$$\mathrm{The}\;\mathrm{disease}\;\mathrm{index}\;\left(\%\right)=\frac{\mathrm{Number}\;\mathrm{of}\;\mathrm{diseased}\;\mathrm{grains}\;\mathrm{in}\;\mathrm{the}\;\mathrm{mutant}}{\mathrm{Number}\;\mathrm{of}\;\mathrm{diseased}\;\mathrm{grains}\;\mathrm{in}\;\mathrm{the}\;\mathrm{wildtype}}\times 100$$


Three sterile filter papers were taken, placed in a glass dish, and moistened with an appropriate amount of ddH_2_O. Using a sterile blade, both ends of four corn silks were cleanly cut and laid flat on the moistened filter papers. An activated mushroom block was inoculated onto the lower end of the corn silk. Five replicates were applied for each strain. The inoculated corn husks were placed in an incubator at a constant temperature of 25 °C for 5–6 days. After incubation, the length of the infection was measured and photographs were taken for record-keeping [39].

## 3. Results

### 3.1. Sequence Analysis of F. graminearum FgPel Gene and Generation of the Gene Mutation

Using SMART and PFAM software, the structural domains of the genes were predicted. The analysis revealed that FGSG_03121 and FGSG_10004 possess the Amb_all domain, while FGSG_10004 and FGSG_13834 have the fungal-type cellulose-binding domain (Figure 1A). The three proteins, FGSG_03121, FGSG_10004, and FGSG_13834, were annotated as hypothetical (or unnamed) proteins in the NCBI database. Protein structural domain analysis revealed that the three genes possessed two pectin cleavage enzyme structural domains, thus categorizing them within the *F. graminearum* pectin cleavage gene family. For convenience, the three proteins were designated as *FgPel1*, *FgPel2*, and *FgPel3*, respectively.

Phylogenetic analysis was performed using MEGA 11.0 software with the following species names and germplasm numbers: *F. graminearum* (XP_011322248.1), *F. spinosum* (AAC64368.1), *F. proliferatum* (KAG4262999.1), *F. albicans* (XP_003711703.1), *F. oxysporum* (PTD03209.1), *Pseudomonas fluorescens Migula* (WP_003173293.1), *F. verticillioides* (SCN79586.1), A. fumigatus (KAH2771735.1), and A. niger (GAQ42273.1). The phylogeny showed that the *FgPel1*, *FgPel2*, and *FgPel3* proteins of *F. graminearum* are evolutionarily conserved (Figure 1B).

### 3.2. Prediction of the Gene Tertiary Structure and Characterization of Physicochemical Properties

The three-dimensional structure of the pectin lyase protein was predicted by I-TASSER (Figure 2A), and pectin lyase genes with the same structural domains had similar tertiary structures. The stereochemistry of the protein structures was verified by Ramachandran plots (Appendix A). For each protein, the server generated five models and quantified the confidence of each model using the C-score. The model with the highest C-score was selected as the best representation of the protein structure. These predictive models were further validated by PROCHECK, confirming the stereochemical quality of the structures. The Psi/Phi Ramachandran plots (Appendix A) showed variability in the backbone residues across different regions, with a low percentage of residues having phi/psi angles in disallowed regions, thus making the model acceptable. The 3D–1D scores from Verify-3D (Appendix A) ranged from 63.97 to 92.53%, indicating that the tertiary structure models were of good quality and correctly folded conformations (Appendix A). In addition, the models were evaluated using the ERRAT server, where ERRAT scores ranged between 80.083 and 84.900, indicating the high quality of the models (Appendix A). The Z-scores calculated by PROSA were −5.99, −2.52, and −4.63, respectively (Figure 2B), which aligned with the NMR/XRD-based structural regions, confirming the reliability of the 3D structural predictions.

The amino acid sequences were analyzed through the online website EXpASY ProtParam tool to obtain various physicochemical characteristics of the proteins (Table 1). The molecular weights of the three proteins ranged from 36,785.35 Da to 79,828.43 Da, while the isoelectric point or theoretical pI values of the pectin lyase proteins of the different strains ranged from 4.25 to 9.03. The isoelectric point (pI) values of *FgPel2* and *FgPel3* were found to be acidic, while that of *FgPel1* was found to be basic. The pH value of the internal environment of an organism is a crucial parameter that may influence the virulence and pathogenicity of the pathogen by affecting the physicochemical properties of protein secretion and expression. Based on the protein instability index, *FgPel1* was a stable protein with an index value below 40. In contrast, *FgPel2* and *FgPel3* were both unstable proteins with indices above 40.

### 3.3. The Generation of Three FgPel Gene Deletion Mutants

To elucidate the functions of *FgPel1*, *FgPel2*, and *FgPel3* in *F. graminearum*, the wild-type (PH-1) strain of *F. graminearum* was transformed by PEG-mediated protoplast. Targeted deletion of the *FgPel* gene was obtained by integrating the thaumatin resistance genes through a transformation process. The molecular validation of *FgPel1*, *FgPel2*, and *FgPel3* is shown in Appendix A. Three independent deletion mutants of *FgPel* were obtained.

### 3.4. The Mutual Regulation of Pectinase Genes FgPel1, FgPel2, and FgPel3 and Their Expression Changes during the Infection Process

RT-qPCR experiments showed that when the *FgPel1* gene was deleted, the expression levels of *FgPel2* and *FgPel3* were downregulated; when the *FgPel2* gene was deleted, the expression level of FgPel3 was upregulated; and when the *FgPel3* gene was deleted, the expression levels of *FgPel1* and *FgPel2* were both upregulated (Appendix A). Prediction from the *F. graminearum* genomic database, FgBase, indicates that gene expression levels increase during pathogen infection. RT-qPCR experiments show that gene expression levels gradually rise at 3, 5, and 7 days post-inoculation (Appendix A). These results suggest that there may be mutual regulation and compensatory mechanisms among *FgPel1*, *FgPel2*, and *FgPel3*.

### 3.5. The Utilization of Different Concentrations of Pectin and Growth Characteristics of Mutant Strains at Different pH Values

To investigate the ability of *FgPel* strains to utilize pectin, the growth of PH-1 and mutant strains on pectin-supplemented PSA medium was analyzed. The colonies of both strains showed irregular morphology (Figure 3A), but the growth of Δ*Fgpel1*, Δ*FgPel2*, and Δ*FgPel3* was inhibited at pectin concentrations of 0.10, 0.50, and 1% compared to PH-1. The growth rate of Δ*Fgpel1* was significantly more inhibited (Figure 3C). These results suggest that the specific gene deletion affected the pectin utilization in *F. graminearum*.

To investigate whether the growth of *FgPel* strains is affected by external acidic and alkaline environments, PH-1 and mutant strains were inoculated on PSA medium with pH values of 4, 6.5, and 8 for 3 days at 25 °C in darkness. The inhibition of mycelial growth of the Δ*FgPel1* strain was more pronounced at pH 4. The best growth for both PH-1 and mutant strains occurred at pH 6.5. At pH 8, the mycelial growth of the Δ*FgPel2*, and Δ*FgPel3* strains was more inhibited (Figure 3B,D).

### 3.6. Pectin Cleavage Genes Are Not Involved in Mycelial and Conidia Growth

To investigate the role of the *F. graminearum* pectin lyase gene in nutrient growth, the strains were cultured on PSA plates for 3 days and statistically analyzed for the colony diameter size and morphology (Figure 4A,B). The mutant strain had a normal nutrient utilization compared with PH-1. The pigmentation of the aerial mycelium was not significantly different from that of PH-1, and there was no significant difference in the colony size. The deletion of the pectin lyase gene does not affect colony growth, conidia formation, or spore germination. These results indicate that the deletion of the pectin lyase gene had no impact on the growth of *F. graminearum*.

*F. graminearum* produces asexual conidia to infect wheat ears during the flowering stage. We investigated the asexual reproduction of the gene deletion mutant and found that the conidia formation and spore-producing ability were not reduced (Table 2). Spore morphology was examined using a Nikon Eclipse Ni-U biomicroscope equipped with differential interference microscopy (Figure 4C). The images showed that the mutant strains produced sickle-shaped conidia with both foot and apex cells, with no significant differences in the conidia morphology. Sporulation of *F. graminearum* was induced in CMC medium, and the resultant spore suspension was transferred to YEPD medium with shaking incubation. After 6 h of germination, the microscopic observations confirmed the consistent germination morphology and rates of the mutant strain, which exhibited growth comparable to that of the PH-1 strain (Figure 4C). These findings indicate that the gene deletion did not significantly affect the mycelial growth, spore morphology, and spore production ability.

### 3.7. Effect of the Gene Deletion Mutants on F. graminearum Virulence

To investigate the role of the *FgPel* gene in the pathogenicity of *F. graminearum*, experiments were conducted to assess the effect of Δ*FgPel* on maize silks and wheat ears. PH-1 and mutants Δ*FgPel1*, Δ*FgPel2*, and Δ*FgPel3* were inoculated at the base of maize silks. The pathogenicity of these mutants was reduced by 29, 45, and 32%, respectively, compared to PH-1, indicating that the gene deletion reduces the pathogenicity of *F. graminearum* (Figure 5A,B). To further evaluate the effect on wheat, inoculation experiments were performed at the flowering stage. Ten microliters of spore suspension (1 × 10^5^ spores/mL) were inoculated into the middle kernels of wheat spikelets. Fourteen days after inoculation, symptoms of the erysipelas infection were observed. Wheat spikes inoculated with the mutant strains showed a significantly reduced pathogenicity, compared to those inoculated with PH-1, while Δ*FgPel1*, Δ*FgPel2*, and Δ*FgPel3* showed reductions in pathogenicity of 23, 52, and 32%, respectively. The deletion of Δ*FgPel2* shows a more pronounced reduction in pathogenicity on wheat spikes. These results indicated that deletion of these genes affected the pathogenicity of *F. graminearum* (Figure 5C,D).

## 4. Discussion

The plant cell wall acts as a natural barrier to limit the pathogen invasion [40]. To invade the host plants, pathogenic fungi produce a variety of cell wall-degrading enzymes. These enzymes disrupt the structure of the plant cell wall by breaking down its key components such as cellulose, pectin, and lignin. Pectin in the cell wall is essential for maintaining structural integrity, and pectinolytic enzymes play a critical role in its degradation. These enzymes are particularly important during fungal infections and have been extensively studied in recent years [41]. In this study, three pectinolytic genes in *F. graminearum* essential for inducing cell death were identified. Our data provided new insights into the host–pathogen interactions.

Structural domain analysis revealed that all three pectin lyase genes had pectin lyase domains. The gene knockouts for *FgPel1*, *FgPel2*, and *FgPel3* were obtained through gene knockout. RT-qPCR experiments showed that the deletion of any one of these genes affects the expression of the other two, indicating potential regulatory and compensatory mechanisms among *FgPel1*, *FgPel2*, and *FgPel3*. This complex regulatory network may be part of the plant’s response to pathogen infection, with the specific mechanisms requiring further investigation. Evolutionary tree analysis showed that these genes were conserved during evolution. I-TASSER tertiary structure analysis showed that pectin lyases with similar domains had comparable tertiary structures. Analysis of the physicochemical properties showed that the pI values of *FgPel2* and *FgPel3* were acidic, while that of *FgPel1* was alkaline. The pH of the internal environment of an organism is a key factor that can influence the virulence and pathogenicity of pathogens by affecting the physicochemical properties of protein secretion and expression [42].

Many studies have shown that pectin lyase genes do not directly affect the growth and development of pathogenic fungi, but play a crucial role in their utilization of pectin. Deletion of the pectin lyase gene in *F. sacchari PL* did not affect spore morphology and production in *F. graminearum*. However, the rate of mycelial growth on pectin agar was significantly reduced and fewer aerial hyphae were produced [43]. Compared with wild-type *F. graminearum*, the *FgPel* knockout mutant exhibited a significantly lower mycelial growth rate and fewer aerial mycelia on the pectin agar, although their spore morphology and production remained similar. This suggests that the *FgPel* gene is involved in the trophic growth and pectin degradation by pathogenic fungi, but not in the regulation of spore formation.

Changes in the external pH environment have been shown to affect the expression of fungal pectin lyase genes. Mutations in the pectin lyase *PelA* of *A. niger* do not affect growth on polygalacturonic acid, but may affect growth and pathogenicity under high pH conditions [44]. In our study, deletion of the *FgPel* gene in *F. graminearum* did not affect its vegetative growth, conidia morphology, or sporulation. However, mycelial growth of the Δ*FgPel1* strain was more inhibited at pH 4. Optimal growth for both wild-type PH-1 and mutant strains was observed at pH 6.5. The Δ*FgPel2* and Δ*FgPel3* strains showed more inhibited colony growth at pH 8.

Studies have shown that deletion of pectin lyase genes reduces the pathogenicity of certain fungi. In the lychee frost blight fungus *Peronophythora litchii*, deletion of the pectin lyase genes plpel1 and *plpel1-like* revealed that the protein LcPIP1 interacts with *PlPeL1* proteins and reduces the virulence of the pathogen [45]. Similarly, deletion of the pectin lyase gene *VdPL1-4* in *V. dahliae* did not affect its spore morphology or production [46]. RT-qPCR experiments show that the gene expression levels gradually rise at 3, 5, and 7 days post-inoculation. Subsequent pathogenicity experiments determined that the pathogenic outcomes of these mutant strains varied between corn silk and wheat ears. Through pathogenicity experiments, this study established that the pathogenic outcomes of these mutant strains varied between corn silk and wheat ear. The deletion of the *FgPel1*, *FgPel2*, and *FgPel3* genes reduced the pathogenicity of *F. graminearum* on corn silk. These results indicate that the three genes have significant functional similarity. Therefore, they may enhance overall pathogenicity through synergistic action during the pathogenic process. In wheat ears, the deletion of these genes also led to a decline in pathogenicity, with the Δ*Fgpel2* mutant showing a more notable pathogenic ability. This suggests that different pectin lyase genes play distinct roles in the infection process of *F. graminearum*, with some acting on the cell wall and others functioning as virulence factors.

In addition, pre-laboratory studies showed that deletion of the *FgPel2* and *FgPel3* genes affected the number and size of *F. graminearum* cysts, but did not have a significant effect on cyst spore morphology. We speculate that these pectin lyase genes may be involved in ascospore formation, and their specific mechanisms of action require further investigation. Deletion of these genes did not affect the vegetative growth of *F. graminearum* but affected its pathogenicity. The functions of different pectin lyase genes vary considerably. Our study provides theoretical support for understanding the role of pectin lyase genes in the growth, development, and pathogenicity of *F. graminearum,* and for developing strategies to prevent and control wheat blight.

## Figures and Tables

**Figure 1 jof-10-00666-f001:**
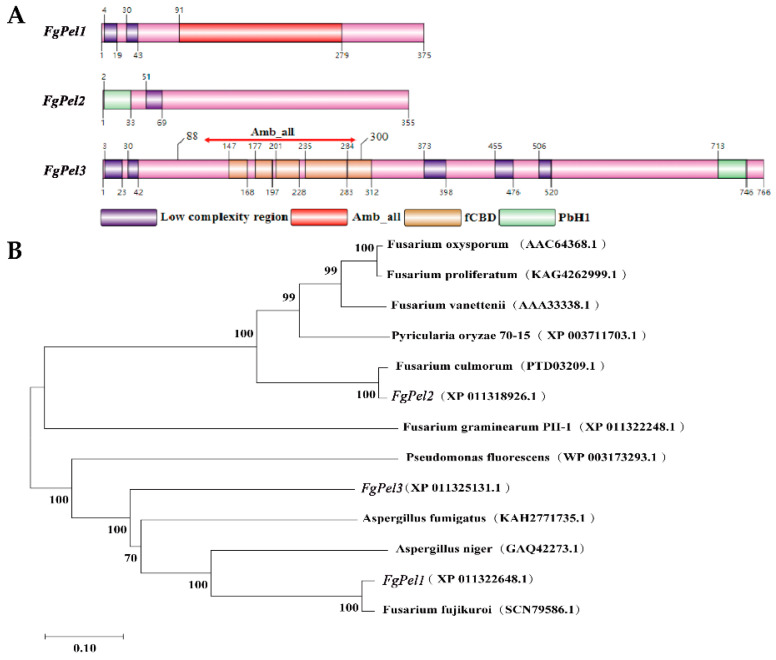
A structural analysis of the *FgPel* gene, encoding a pectin cleavage enzyme in *F. graminearum,* was conducted. (**A**) Structural domain maps of the three genes, *FgPel1*, *FgPel2*, and *FgPel3* were generated using IBS2.0. The domains are as follows: L Low complexity region: a region of low compositional complexity; Amb_all: a domain associated with the pectate lyase family; Fcbd: a fungal-type cellulose-binding domain; and PbH1 domain: parallel beta-helix repeats. (**B**) A phylogenetic tree and structural domain sequence analysis of different species were constructed with Mega11.0. A minimal phylogenetic tree was generated by comparing the full-length sequences of *FgPel* homologous proteins across different fungi. This tree was constructed using the Mega11.0 software with a minimal evolution algorithm and 1000 bootstrap replicates. The scale bar indicates the number of substitutions per site, and the number next to each species name corresponds to the NCBI protein accession number.

**Figure 2 jof-10-00666-f002:**
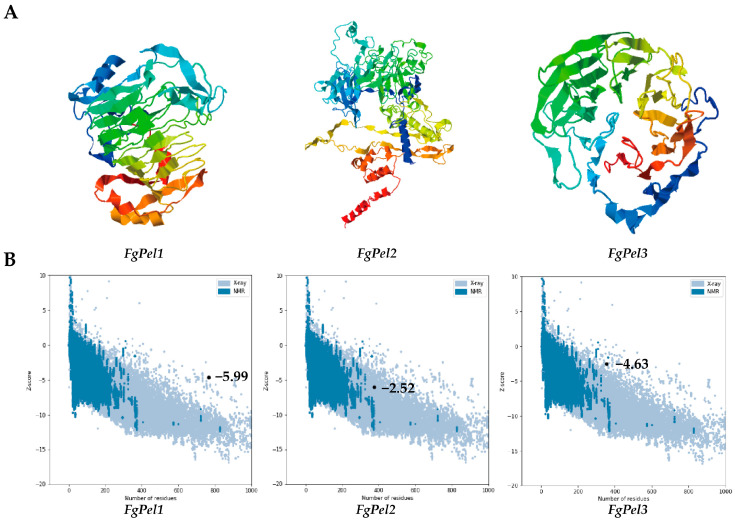
Three-dimensional structures of pectin lyase. (**A**) The tertiary structures of the three pectin lyase genes were predicted by the I-TASSER software. From left to right, they are *FgPel1*, *FgPel2*, and *FgPel3*. (**B**) PROSA plot showing Z-score of −5.99, −2.52, and −4.63 for the I-TASSER.

**Figure 3 jof-10-00666-f003:**
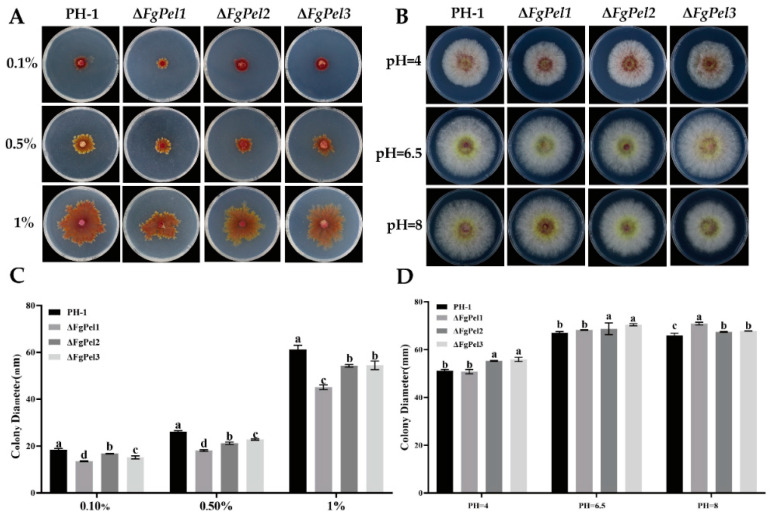
Effect of *FgPel* mutants on pectin utilization and growth of *F. graminearum* under different acidic and alkaline conditions. (**A**) Comparison of the growth of PH-1, Δ*FgPel1*, Δ*FgPel2*, and Δ*FgPel3* on pectin plates at 0.1, 0.5, and 1% after 5 days of incubation at 25 °C. (**B**) Colony morphology of PH-1, Δ*FgPel1*, Δ*FgPel2*, and Δ*FgPel3* under different pH conditions after 5 days of incubation. (**C**) Inhibition rate of mycelial growth of PH-1 and mutants by different pectin concentrations. (**D**) Inhibition rate of mycelial growth of PH-1 and mutants under different pH conditions. Different letters on the bars indicate significant differences at *p* ≤ 0.05 by Duncan’s multiple range test. Values followed by the same letter indicate no significant difference. All experiments were repeated three times.

**Figure 4 jof-10-00666-f004:**
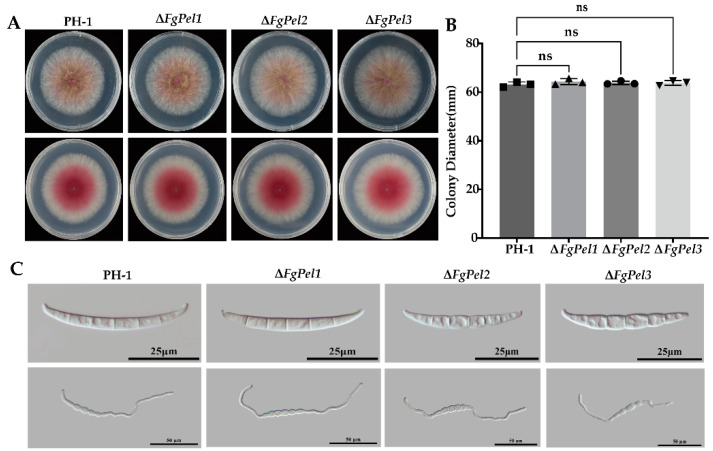
Observation of the colony and conidia morphology of PH-1 and *FgPel* gene mutant strains. (**A**) Colony morphology of PH-1 and *FgPel* gene deletion mutant strains after incubation on PSA at 25 °C for 3 days, respectively. After culturing in YEPD liquid for 6 h, the morphology of spore germination of PH-1 and *FgPel* gene deletion mutants was observed under microscope. (**B**) Colony diameters of these strains counted after 3 days of growth. (**C**) Conidia morphology of PH-1 and *FgPel* gene deletion mutant strains was observed under the microscope after 3 days of culturing in CMC liquid medium. The ‘ns’ on the bar graph indicates that there is no significant difference at *p* ≤ 0.05 according to Duncan’s multiple range test. All experiments were replicated three times in three replications.

**Figure 5 jof-10-00666-f005:**
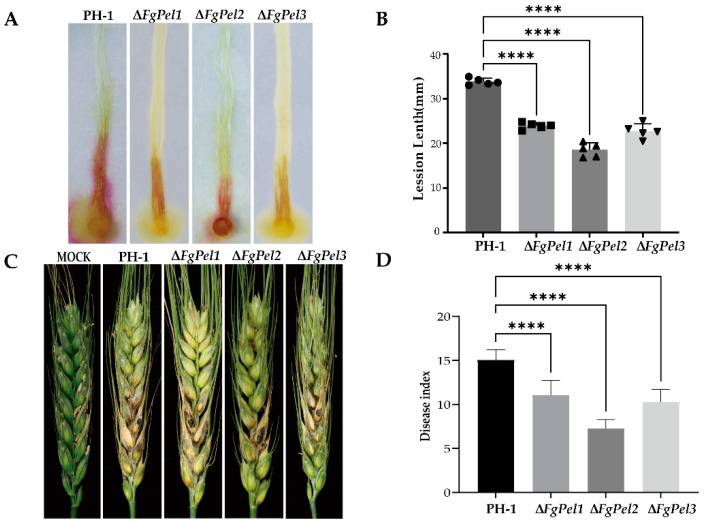
Virulence of PH-1 and *FgPel* gene deletion mutants on corn silks and wheat ears. (**A**) Brown necrosis caused by inoculation with PH-1, ∆*FgPel1*, ∆*FgPel2*, and ∆*FgPel3* strains on corn silks. Pictures were taken after 5 days of incubation at 25 °C. (**B**) Disease symptoms on flowering wheat spikes after injection of conidial suspension (10 µL, 1 × 10^5^ conidia/mL) of PH-1, ∆*FgPel1*, ∆*FgPel2*, and ∆*FgPel3* strains. (**C**) Length of the necrotic lesions on cornhusks caused by each strain, measured 14 days post-inoculation.The different shaped symbols on the bar chart represent different experimental groups, and the experiment was repeated five times. (****) on the bars indicates a significant difference at *p* ≤ 0.01 according to *t*-test. (**D**) Disease index of F. graminearum was determined at 14 dpi. A minimum of 20 wheat ears were inoculated per replication. Error bars represent the standard error of the mean. (****) on the bars indicates a significant difference at *p* ≤ 0.01 according to *t*-test.

**Table 1 jof-10-00666-t001:** Physicochemical properties of the pectin cleavage genes.

Strain	Accession Numbers	Number of Amino Acids	SP (aa)	Molecular Weight	Isoelectric Point (PI)	Instability Index	Fat Factor	Relative Average Hydrophilicity (GRAY)	Total Anionic Residue Number (Asp + Glu)	Total Cationic Residue (Arg + Lys)
*FgPel1*	NC_026475.1	375	1–18	39,816.88	9.03	24.21	78.03	−0.171	25	33
*FgPel2*	NC_026474.1	355	NO	36,785.35	4.27	41.25	58.54	−0.386	49	23
*FgPel3*	NC_026476.1	766	1–18	79,828.43	4.25	46.73	55.52	−0.592	134	55

**Table 2 jof-10-00666-t002:** Conidia production of the wild-type *F. graminearum* PH-1 and *FgPel* mutants.

Strains	Conidiation (10^6^ Conidia/mL)
PH-1	1.04 ± 0.12 ^a^
Δ*FgPel1*	1.19 ± 0.16 ^a^
Δ*FgPel2*	0.92 ± 0.01 ^a^
Δ*FgPel3*	1.04 ± 0.22 ^a^

Mean ± SE was calculated from the results of three independent experiments. Values on the table followed by the same letter are not significantly different at *p* ≤ 0.05 according to Duncan’s multiple range test. Conidia production was assessed in 100 mL of carboxymethylcellulose (CMC) medium after 5 days of incubation at 25 °C and 150 rpm. Each experiment was replicated three times, with three replicates per experiment.

## Data Availability

The data presented in this study are available on request from the corresponding author.

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
