# Peer review of "Roles of Three FgPel Genes in the Development and Pathogenicity Regulation of Fusarium graminearum"

_jof, 2024, doi:10.3390/jof10100666_

Round 1
Reviewer 1 Report
In general, the manuscript is well written and the experiments were perfectly performed, but
1. The restoration of the phenotype of each deltant strain by the reintroduction of the corresponding Wt gene has been missing all along of the experiments.
2. is it possible that the deletion of one of the FfPel gene could alter the transcription of the other two genes? these experiment is also missing in the results.
3. is it possible for the three different genes to have redundant functions? the proper experiments are also missing
4. When are the three FgPal genes transcribed?
line 22. correct “concentrationsm”
line 22. “with the inhibition of what? reprase it to make it clear
Line 29. correct “exter-nal”
Line 30. correct “F. graminearumwithout”
Line 37. correct “[1-3].F. ”
The different domains represented with different colors in Figure 1 Panel A, are not described in the text or the figure legend. Are these putative enzymes secreted? if so, indicate the signal motif.
line 236. “27932.72 to 79828.43 Da,” check with those MW of Table 1.
Figure 4. In c and D, the font sizes of the name of the strains make them bigger for a better visualization.
Panel B. in which media?
Line 288. “(Fig. 2A, B)”. could not find the corresponding figures. check
Lines 292-293. These results indicate that deletion of the pectin lyase gene had no effect on the growth of 292 F. graminearum. be specific under which conditions
Figure 5, Panel C. are two sets of photos, taht are not properly described in figure legend.
Figure 4. In c and D, the font sizes of the name of the strains make them bigger for a better visualization.
Panel B. in which media?
Reviewer 2 Report
The authors selected and analyzed three pectin lyase genes (Fgpel1, 2, 3) of Fusarium graminearum and generated mutants. The gene deletion mutants affected the morphology and growth rate of F. graminearum in the media containing pectin. Pathogenicity assays demonstrated that deletion mutants significantly reduced the ability of F. graminearum to infest corn silks and wheat ears.
The manuscript presented some new information on identification and characterization of F. graminearum pectin lyase genes. My major concern is that there are no complementation experiments for the deletion mutants. It is needed to conduct complementation to confirm the function of three pectin lyase genes for publication.
Other minor points:
Please clarify you are working on wheat blight not wheat blast. These diseases are caused by different fungal pathogens. Line 46 and others
Line 45-49, why are you focus on Aspergillus instead of Fusarium? There are multiple reports on Fusarium fungicide resistant strains in the literatures.
Fig.1A, I am surprised there are no signal peptides are labeled? All theses proteins are secreted proteins?
Table 1 Number of amino acids from mature protein or including signal peptides?
Line 45: 20 mL of pectin medium, please clarify what you used. Liquid media or agar plate. From Fig.4A, it appears you used agar plate. What else in the medium? Need some details.
Fig. 4A should include a control without pectin.
I do not think Fig. 3 and 4 provide any real useful information. They should be in supplementary.
Fig. 6: Toxin data should be provided.
see above
Author Response
Reviewer 1:
|
|||||||||||||||||||||||||||||||||||||||||||||||||||||||||||||||||||||||||||||||||
Round 2
Reviewer 1 Report
I am agree with the changes of this new version of the manuscript. In this sense I accept the manuscript for its publication.
No comments
Author Response
Thank you very much for your attentive concern and valuable suggestions. We have carefully revised and improved the article. Your opinions are crucial to us. Thank you again for your support and assistance!
Reviewer 2 Report
The authors resolved some of my concerns but did not do the complementation experiments to confirm the gene functions.
The authors resolved some my concerns but did not do the complementation experiments to confirm the gene functions.
Author Response
Please see the attachment, thanks.

Round 3
Reviewer 2 Report
The revised version added double mutants, still did not do complementation.
Without gene complementation, I do not think it is a complete story.
The revised version added double mutants, still did not do complementation.
Without gene complementation, I do not think it is a complete story.
